# Transcriptome Analysis of Apples in High-Temperature Treatments Reveals a Role of MdLBD37 in the Inhibition of Anthocyanin Accumulation

**DOI:** 10.3390/ijms23073766

**Published:** 2022-03-29

**Authors:** Yu-Feng Bu, Shuo Wang, Chen-Zhiyu Li, Yue Fang, Ya Zhang, Qing-Yu Li, Hai-Bo Wang, Xue-Sen Chen, Shou-Qian Feng

**Affiliations:** 1National Key Laboratory of Crop Biology, National Research Center for Apple Engineering and Technology, College of Horticulture Science and Engineering, Shandong Agricultural University, Tai’an 271018, China; buqian520@126.com (Y.-F.B.); w15275382665@163.com (S.W.); 17863807683@163.com (C.-Z.L.); 17861510265@163.com (Y.F.); zhangya829829@163.com (Y.Z.); 2Yantai Academy of Agricultural Science, Yantai 264000, China; liqingyu613891@163.com; 3Shandong Institute of Pomology, Tai’an 271000, China; wanghaibo992@126.com

**Keywords:** apple, high temperature, anthocyanin, *MdLBD37*

## Abstract

Coloring in apple fruit due to anthocyanin accumulation is inhibited by high temperature; however, the underlying mechanism remains unclear. In the present study, total anthocyanin and cyanidin 3-galactoside contents were determined and compared between cv. ‘Redchief Delicious’ apple fruits at 25 °C and 35 °C treatments. The high temperature (35 °C) treatment substantially decreased total anthocyanin and cyanidin 3-galactoside contents. The transcriptomes of 25 °C- and 35 °C-treated apples were analyzed by high-throughput RNA sequencing. A total of 8354 differentially expressed genes (DEGs) were detected at four time points corresponding to the two temperature treatments. The up-regulated DEGs were annotated using GO as well as KEGG databases. A network module of 528 genes (including 21 transcription factors) most associated with the total anthocyanin and cyanidin 3-galactoside contents was constructed by weighted correlation network analysis (WGCNA). In the WGCNA module, we unearthed a LOB domain-containing gene designated as *MdLBD37*. The expression of *MdLBD37* was sharply up-regulated by high temperature and negatively correlated with the total anthocyanin and cyanidin 3-galactoside contents. Overexpression of *MdLBD37* in apple fruit and calli decreased the expression of anthocyanin biosynthetic genes, such as *MdCHI*, *MdCHS*, *MdF3H*, *MdANS*, *MdDFR*, and *MdUFGT*, along with anthocyanin accumulation. Our results suggested that *MdLBD37* significantly influenced the high-temperature inhibition of anthocyanin accumulation in apples. The findings shed more light on the mechanism of anthocyanin inhibition during high-temperature stress in apples.

## 1. Introduction

Color is an important index to measure the appearance quality of an apple (*Malus domestica* Borkh.), which largely determines its commodity value and market popularity [1]. Fruit color is closely related to the anthocyanin components and contents in the pericarp. The anthocyanin content in the peel directly affects fruit coloring and usually differs across the variety [2,3,4]. Generally, the anthocyanin content of a variety is mainly determined by genetic factors [5], but environmental factors, including temperature, are other important regulators [4,6,7].

HT (high temperature) inhibits anthocyanin accumulation in apples, affecting fruit coloration and economic value [8,9,10]. The accelerated urbanization and industrialization, and the corresponding release of greenhouse gases, have boosted global warming [11]. In recent years, the frequency of high-temperature weather has especially increased, thereby affecting larger areas [12]. Continuous high-temperature weather during the summer and autumn is making the phenomenon of poor apple coloration increasingly serious [8]. As a result, unraveling the molecular foundation of high-temperature inhibition of apple coloring is crucial.

Several studies reported that high temperature inhibits anthocyanin synthesis by reducing the activity of associated enzymes and the expression of anthocyanin-related genes [13,14]. Thus far, anthocyanin biosynthetic genes, including *PAL* (phenylalanine ammonia-lyase), *CHS* (chalcone synthase), *CHI* (chalcone isomerase), *F3H* (flavanone 3-hydroxylase), *DFR* (dihydroflavonol 4-reductase), and *UFGT* (uridine diphosphate-glucose: flavonoid 3-O-glucosyltransferase), have been isolated and identified from apple [4,15,16]. The expression of these structural genes is regulated by different kinds of transcription factors (TFs), such as MYB, bHLH, LBD, NAC, and WD40 [17,18,19,20,21]. *MdMYB1* was identified as the key gene which regulates anthocyanin synthesis in apple peel [22]. Recent studies have found that the expression of *MdMYB1*, *MdCHI*, *MdF3H*, and *MdUFGT* was inhibited by high temperatures [8,9]. However, the regulatory mechanism of high-temperature inhibition of anthocyanin accumulation in apples has not been fully understood.

In the present study, the total anthocyanin and cyanidin 3-galactoside contents and transcriptome in the peel of ‘Redchief Delicious’ apples treated at 25 °C and 35 °C were estimated and compared to understand the mechanism of high-temperature inhibition of apple coloration. 

## 2. Results

### 2.1. Decrease in Anthocyanin Accumulation in Apple Due to HT Treatment

The coloration of ‘Redchief Delicious’ apple fruit was markedly reduced by the HT (35 °C) treatment; hence, no obvious color changes were noted on the fruit peel throughout the treatment period. In contrast, red coloration of the 25 °C-treated apple fruit was first detected 3 days after initiating the treatment, following which the color continued to intensify until the fruit peel was completely red (Figure 1A). Consistent with the differences in coloration, corresponding contents of total anthocyanin and cy-3-gala continued to increase in the 25 °C-treated fruit over the treatment period, whereas they steadily decreased in the 35 °C-treated fruit (Figure 1B,C). Thus, high-temperature exposure repressed anthocyanin synthesis in the apple fruit.

### 2.2. Gene Expression Differences between RT- and HT-Treated Apples

RNA-seq analysis of the 25 °C- and 35 °C-treated apple fruits (collected at four time points) generated a total of 153.1 Gb clean data with Q30 > 95%. The clean reads were mapped into reference genome sequences of apple (Appendix A). The gene transcript levels were quantified by the RPKM values. A total of 8354 DEGs were detected from the four pairwise comparisons (absolute log2 fold change > 1, adjusted *p* < 0.05). Specifically, 3934, 3160, 2582, and 2752 DEGs were identified at T1, T2, T3, and T4 (3, 6, 9, and 12 days after temperature treatment) between 25 °C- and 35 °C-treated fruits (Appendix A). Out of these, 1505, 1806, 1090, and 1781 genes (at T1, T2, T3, and T4, respectively) had down-regulated expression levels, while 2429, 1354, 1492, and 971 genes (corresponding to T1, T2, T3, and T4, respectively) showed up-regulated expression levels (Figure 2A). A Venn diagram presenting the distribution of DEGs revealed 344 DEGs that were common to all four comparisons. Additionally, 2021, 1189, 1226, and 1234 DEGs were specific to the 35 °C vs. 25 °C comparisons at T1, T2, T3, and T4, respectively (Figure 2B).

To assess the reliability of the RNA-seq data, 10 DEGs were chosen for a qRT-PCR assay. The trends in the expression-level changes determined by the qRT-PCR analysis were consistent with those in the RNA-seq data (Figure 2C). Thus, the RNA-seq data were confirmed as highly reliable and useful for identifying the functional genes in apple.

### 2.3. Functional Classification of DEGs

Most of the functionally annotated (via GO analysis) DEGs were associated with ‘phosphoric ester hydrolase activity’, ‘nucleus’, and ‘membrane-bounded organelle’ in the 35 °C vs. 25 °C comparisons at T1. At T2, most of the annotated genes were related to ‘catalytic activity’ in the 35 °C vs. 25 °C comparisons. In comparison, most of the DEGs were related to the ‘photosystem’ and ‘membrane’ in the 35 °C vs. 25 °C comparisons at T3 and T4, respectively (Figure 3). The KEGG results indicated that carbon metabolism was the most significantly enriched pathway of DEGs in the 35 °C vs. 25 °C comparisons at T1, T2, T3, and T4 (Figure 4). 

### 2.4. Anthocyanin-Related DEGs Revealed by Analysis of Co-Expression Networks

A total of eleven WGCNA modules were identified in the analysis of co-expression networks (Figure 5A,B). Interestingly, the ‘green’ module comprising of 528 genes was most highly correlated with the total anthocyanin and cy-3-gala contents (Appendix A). Twelve genes in the ‘green’ module were anthocyanin structural genes (encoding six *CHSs*, two *F3Hs*, one *DFR*, and two *LDOXs*), whereas the remaining genes were novel. The structural genes showed high expression within apple fruits at 25 °C treatment compared to those at 35 °C treatment (Figure 5D). 

There were 21 TF genes in the ‘green’ module (Appendix A). Most of these TF genes showed a down-regulation tendency at the 35 °C-treated apples compared to the 25 °C-treated fruits. However, homeodomain-like superfamily protein (MD02G1166700) and *MdLBD37* (MD15G1294700) showed obvious up-regulated expressions in the 35 °C-treated apples compared to the 25 °C-treated fruits (Figure 5D). 

Furthermore, we performed correlation analyses using the above 21 TF genes and 12 anthocyanin structural genes. A total of 98 notable correlations (correlation coefficient, R2 > 0.9) were detected between 15 TF genes and 12 anthocyanin structural genes. The results indicated that these TF genes might be important candidate anthocyanin regulators. Most of these TF genes showed positive correlations with anthocyanin structural genes, while homeodomain-like superfamily protein (MD02G1166700) and *MdLBD37* (MD15G1294700) showed negative correlations with anthocyanin structural genes (Figure 5C). Next, we chose *MdLBD37* to test its function in regulating anthocyanin synthesis in apples.

### 2.5. MdLBD37-Mediated Inhibition of Anthocyanin Synthesis in Apple

We injected the plasmids *MdLBD37*-TRV and *MdLBD37*-pRI into the pericarps of young (50 DAFB) and coloring (140 DAFB) fruits of ‘Otome’, respectively. Compared with the control, the transient silencing of *MdLBD37* significantly induced the accumulation of anthocyanins around the injection site (Figure 6A–C). Consistently, anthocyanin biosynthetic genes, including *MdCHI*, *MdCHS*, *MdF3H*, *MdDFR*, and *MdUFGT*, were up-regulated around the injection site. On the contrary, overexpression of *MdLBD37* significantly inhibited the accumulation of anthocyanins and the expression of *MdCHI*, *MdCHS*, *MdF3H*, *MdDFR*, and *MdUFGT* around the injection site (Figure 6D). This observation suggested that *MdLBD37* inhibited anthocyanin synthesis in apples.

Furthermore, we also generated transgenic red-fleshed apple calli overexpressing *MdLBD37* (*MdLBD37*-OE). The overexpression of *MdLBD37* in red-fleshed calli weakened the calli red color and lowered the expression levels of anthocyanin biosynthetic genes *MdCHI*, *MdCHS*, *MdF3H*, *MdDFR*, and *MdUFGT* (Figure 6E–H). These results indicate that *MdLBD37* might repress anthocyanin synthesis by inhibiting anthocyanin biosynthetic genes.

## 3. Discussion

Several studies have reported that fruit red color formation is weakened by a relatively higher ambient temperature [8,14,23]. Similarly, we found that high temperature significantly inhibited anthocyanin accumulation, resulting in the poor coloring of apples. Nonetheless, further investigations are required to illustrate the detailed mechanism of anthocyanin inhibition in apples at high-temperature stress. Transcriptome analysis of high-temperature stressed apple fruits was conducted in the present study to identify the genes involved in regulating anthocyanin production.

Conjoint analysis of transcriptome data by WGCNA can be used for finding the key genes responsible for the phenotypic difference. Recently, genes involved in anthocyanin-deficient yellow-skin somatic mutation in apples have been analyzed using WGCNA, and *MdMYB10* was successfully detected as the candidate gene [24]. The present study identified one module (green) of 528 genes that significantly correlated with the total anthocyanin and cy-3-gala contents using the WGCNA method. The spatiotemporal difference of anthocyanin accumulation is dependent on the structural genes. In the ‘green’ gene module, 12 anthocyanin structural genes, including six *CHSs*, two *F3Hs*, one *DFR*, and two *LDOXs*, were identified. It was evident that the expression of these anthocyanin structural genes was positively related to the total anthocyanin and cy-3-gala contents. Compared to the RT-treated apples, the expression of these anthocyanin structural genes in HT-treated apples was significantly lower, indicating their key role in the high-temperature inhibition of anthocyanin biosynthesis. The observations are similar to the former studies showing that high temperature decreased anthocyanin accumulation by inhibiting anthocyanin synthesis-related enzyme activity and coding gene expression [8,9,14]. 

Negative regulatory genes exert a vital function in inhibiting anthocyanin synthesis in plants at high temperatures. For example, *MdCOL4* was found to inhibit anthocyanin synthesis in apples at high temperatures [9]. In potatoes, the reduction of anthocyanin accumulation was caused by enhancing *StMYB44* expression [25]. In our study, *MdLBD37* in the ‘green’ gene module was identified as an important anthocyanin-negative gene in apples. We found that *MdLBD37* inhibited anthocyanin synthesis and the expression of anthocyanin structural genes, and it was sharply induced by high temperatures. This indicated that *MdLBD37* might be a new negative regulatory gene in the high-temperature regulation of anthocyanin accumulation in apples. Previously, *AtLBD37*, a homologous gene of *MdLBD37* in *Arabidopsis,* was identified as an anthocyanin negative regulatory gene working together with an MBW complex, and it was up-regulated in response to N/NO^3^–signals [26]. However, the regulatory mechanism of *MdLBD37*-mediated high-temperature signal inhibition of anthocyanin synthesis has not previously been described, and this will be a very interesting topic for future research.

Anthocyanin synthesis is transcriptionally regulated by a set of TFs [27,28]. In the ‘green’ gene module, we found different TFs, including ERF, LBD, GRAS, G2-like, Dof, NAC, HD-ZIP, DBB, Trihelix, NF-YC, HSF, and MYB related members, some of which had been previously shown to regulate anthocyanin synthesis. For example, *MdMYB1*, encoding an R2R3-MYB TF, was necessary for anthocyanin synthesis in apple skin [22]. In addition, it was shown that HT inhibited the *MdMYB1* gene. *MdERF1B* and *MdERF3*, encoding ERF TFs, and *MdNAC029*, encoding an NAC TF, have been shown to promote anthocyanin synthesis in apples [29,30,31]. It could be possible that TFs identified in the ‘green’ gene module might participate in regulating high-temperature effects on anthocyanin accumulation in apples.

Above all, anthocyanin biosynthesis in apples is inhibited at HT, mainly at the transcriptional level [7]. Conjoint analysis of transcriptome data by WGCNA revealed a module (green) that exhibited a significant correlation with the total anthocyanin and cy-3-gala contents. *MdLBD37* in the ‘green’ gene module might be an important negative regulator located at the upstream of anthocyanin structural genes and play an important inhibitory role in anthocyanin accumulation at high temperatures (Figure 7). The results, therefore, have important implications for the molecular mechanism of anthocyanin regulation at high temperatures. We will focus on the functions of LBD TFs and other regulators identified in the WGCNA network in the future.

## 4. Materials and Methods

### 4.1. Collection and Temperature Treatment of the Apple Fruits

The fruits were reaped from apple (*Malus domestica*) cultivar ‘Redchief Delicious’ trees growing in an orchard in Yantai, Shandong, China (37°5′ N, 122°1′ E) at 126 DAFB (days after full bloom), when the peel began coloring. The collected fruits were exposed to 20,000 lx light intensity at 35 °C and 25 °C temperatures. We harvested fruit peels at 3, 6, 9, and 12 days of post-treatment, followed by rapid freezing within liquid nitrogen and preservation under −80 °C till further analysis.

Fruits were harvested from apple (*Malus domestica*) cultivar ‘Otome’ trees growing in the experimental farm of the Shandong Pomology Institute (Taian, Shandong, China, 36°11′ N, 117°61′ E) at 50 and 140 DAFB, for injection assays. Transient silencing and overexpression assays using fruits at 50 and 140 DAFB, respectively. ‘Purple 3’ red-fleshed apple calli were cultured on Murashig and Skoog (MS) medium at 25 °C in the dark. The calli were cultured for 15 days before being used for genetic transformation.

### 4.2. Anthocyanin Assessment

Apple fruit peels were ground in liquid nitrogen. After that, 0.5 g ground sample was subjected to 12 h 1% (*v*/*v*) HCl-methanol (10 mL) extraction in the dark at 4 °C. The resultant extract for each sample was centrifuged for 3-min at 6000× *g* followed by absorbance measurement at 510 and 700 nm using the UV-2450 spectrophotometer (Shimadzu, Kyoto, Japan). Total anthocyanin content was determined using the pH-differential method [32]. The resultant extract was filtered through a 0.2 µm polyethersulfone (PES) filter (Krackeler Scientific, Albany, NY, USA) and analyzed using a high-performance liquid chromatography (HPLC) system (LC2010C, Shimadzu, Japan). Cy-3-gala (cyanidin 3-galactoside) was identified with the HPLC system, followed by detection of its content using an ultraviolet detector at 520 nm and calibration curves derived from a cy-3-gala standard (Sigma-Aldrich, St. Louis, MO, USA). Three independent biological replicates were used for all the analyses.

### 4.3. RNA-Seq and Data Analysis

Total RNA was extracted using TRIzol reagent (Invitrogen, Carlsbad, CA, USA). Total RNA was treated with the RNase-Free DNase Set (Qiagen, Nasdaq, NY, USA). RNA quality was assessed using the Agilent 2100 Bioanalyzer (Agilent Technologies, Palo Alto, CA, USA). This was followed by cDNA preparation based on high-quality RNA. Thereafter, we adopted Illumina HiSeq™ 2000 system to sequence cDNA libraries. Low-quality reads were removed from raw RNA-seq reads. Afterward, clean reads were mapped to build a reference genome sequence for apple. Fragments Per Kilobase of transcript (FPKM) approach was utilized to determine gene expression levels [33]. Differentially expressed genes (DEGs) were detected using the DEGseq R package [34], with an adjusted *p*-value ≤ 0.05 and a fold-change ≥ 2 as the significance thresholds. 

### 4.4. GO and KEGG Enrichment Analysis

DEGs were subjected to Gene Ontology (GO) annotation using the Blast2GO software. The GO terms of those DEGs showing adjusted *p* < 0.05 were significant [35]. We also conducted a Kyoto Encyclopedia of Genes and Genomes (KEGG) analysis of the DEGs in KOBAS software [36,37]. 

### 4.5. WGCNA and Correlation Analyses of Anthocyanin-Related Genes

We built gene co-expression networks based on the DEGs using the weighted correlation network analysis (WGCNA) in the R package [38]. The DEGs from the HT- and RT-treated samples were used for co-expression analysis for each of the four time points. Total anthocyanin and cy-3-gala contents were examined for correlation with the modules and all the genes in each module. Significant anthocyanin-related modules were detected based on the highest correlation values with total anthocyanin as well as cy-3-gala contents (Appendix A).

### 4.6. qRT-PCR Analysis

We performed qRT-PCR with a 20 µL sample in the iCycler iQ5 system (Bio-Rad, Hercules, CA, USA). The reaction mixture contained 10 µL SYBR Green PCR Master Mix (TransGen, Beijing, China), 1 µL respective primer, and 1 µL cDNA. Each qRT-PCR reaction was performed with three biological replicates. The transcripts were analyzed using the (Ct) 2^–ΔΔCt^ method [39]. *MdActin* was used as an internal control. Appendix A presents all the qRT-PCR primers.

### 4.7. Transient Expression of MdLBD37 in Apple Fruit

The transient expression assay was conducted using the apple cultivar ‘Otome’. The CDS and another 393 bp fragment of *MdLBD37* were inserted into the pRI101 and pTRV2 vectors [40]. The recombinant plasmids *MdLBD37*-pRI and *MdLBD37*-pTRV were transformed into *Agrobacterium tumefaciens* GV3101. Fruits of ‘Otome’ collected at 50 DAFB and 140 DAFB were injected with transformed *A. tumefaciens* containing *MdLBD37*-pTRV and *MdLBD37*-pRI, respectively. Five days after injection, the apple peel around the injection site was sampled for phenotypic analysis, anthocyanin content determination, and qRT-PCR assays. Ten successfully infected fruits were considered to determine anthocyanin content and gene expression.

### 4.8. Transformation of the Red-Fleshed Apple Calli with MdLBD37

The CDS of MdLBD37 was ligated into the SalI and SmaI sites of the pRI101 vector to generate the overexpression vector MdLBD37-pRI. The recombinant plasmids were transformed into Agrobacterium tumefaciens LBA4404. ‘Purple 3’ red-fleshed apple calli were used for transformation. Transgenic apple calli were selected on MS medium containing 50 mg L^−1^ kanamycin and 250 mg L^−1^ carbenicillin. Each successfully infected apple calli line is used as a biological repeat to determine anthocyanin content and gene expression.

## 5. Conclusions

In this study, we compared the total anthocyanin and cyanidin 3-galactoside contents and transcriptome in the peel of ‘Redchief Delicious’ apples treated at 25 °C and 35 °C. We identified one module that exhibited a significant correlation with the total anthocyanin and cy-3-gala contents using the WGCNA method. We characterized *MdLBD37* in the module as an important negative regulator in anthocyanin accumulation at high temperatures. Our findings provide useful insights into the molecular mechanism of anthocyanin inhibition at high temperatures in apples, which lays the foundation for the genetic improvement of fruit coloration in apples under high temperatures.

## Figures and Tables

**Figure 1 ijms-23-03766-f001:**
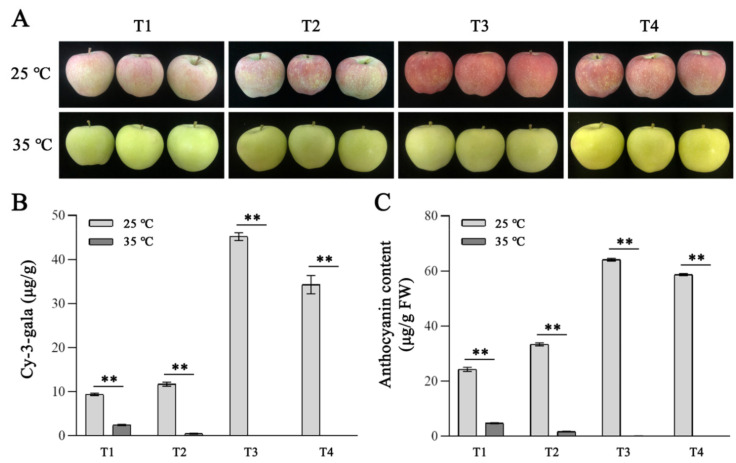
Phenotypes and total anthocyanin and cy-3-gala contents of apples at high-temperature treatment. (**A**) The phenotypes of apples at T1, T2, T3, and T4. (**B**) Cy-3-gala contents of apples at T1, T2, T3, and T4. (**C**) The total anthocyanin contents of apples at T1, T2, T3, and T4. T1, T2, T3, and T4 represent apples after 3, 6, 9, and 12 days of 25 °C and 35 °C treatment, respectively. Values stand for means ± SD of three independent biological replicates. Statistical significance was measured using Student’s *t*-test (** *p* < 0.01).

**Figure 2 ijms-23-03766-f002:**
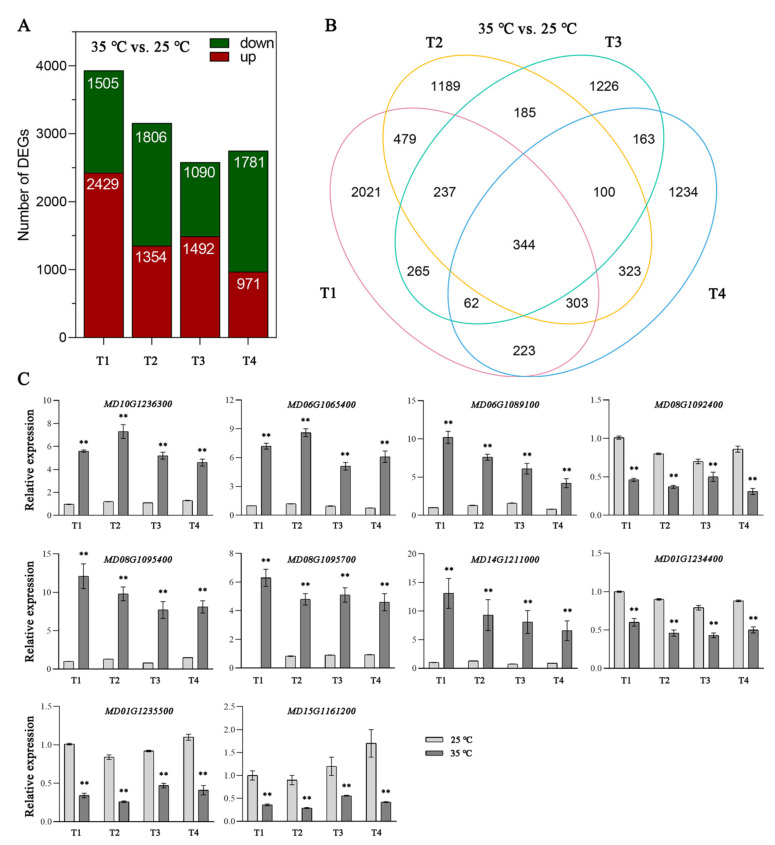
Histogram and Venn diagram showing numbers of differentially expressed genes (DEGs) in the comparisons 35 °C vs. 25 °C at T1, T2, T3, and T4. (**A**) Histogram showing DEGs in the comparisons 35 °C vs. 25 °C at T1, T2, T3, and T4. (**B**) Venn diagram presenting the overlap of DEGs in the comparisons 35 °C vs. 25 °C at T1, T2, T3, and T4. (**C**) Expression levels analysis of 10 selected DEGs by qRT-PCR. T1, T2, T3, and T4 represent apples after 3, 6, 9, and 12 days of 25 °C and 35 °C treatment, respectively. Values stand for means ± SD of three independent biological replicates. Statistical significance was measured using Student’s *t*-test (** *p* < 0.01).

**Figure 3 ijms-23-03766-f003:**
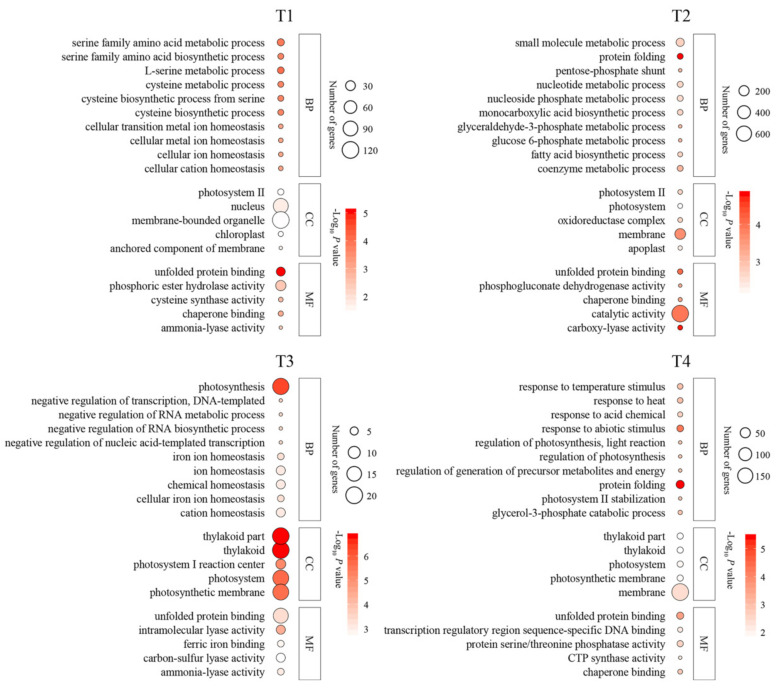
Gene Ontology (GO) enrichment analysis of differentially expressed genes (DEGs) in the apple in the comparisons of 35 °C vs. 25 °C at T1, T2, T3, and T4. The DEGs were presented in three main types, respectively, ‘biological process’, ‘cellular component’, and ‘molecular function’. T1, T2, T3, and T4 represent apples after 3, 6, 9, and 12 days of 25 °C and 35 °C treatment, respectively.

**Figure 4 ijms-23-03766-f004:**
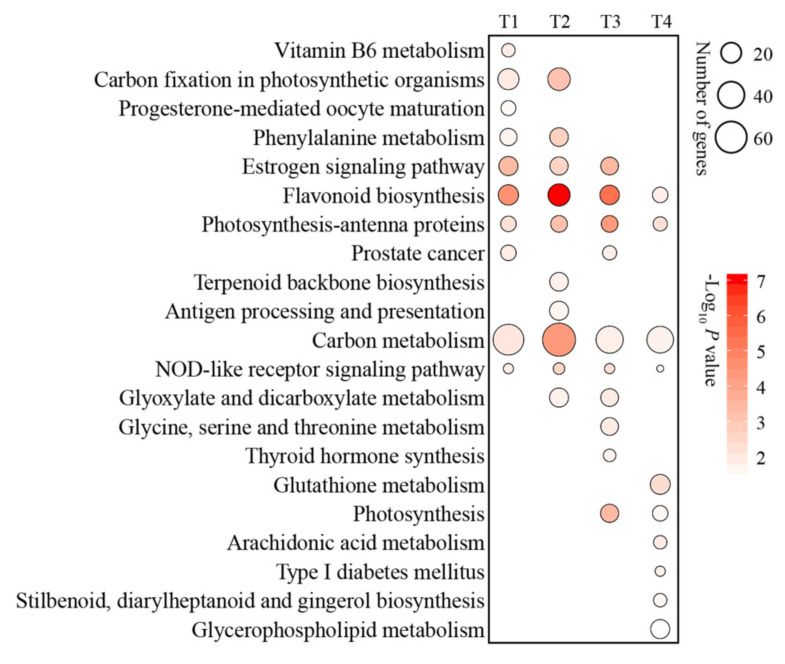
Kyoto Encyclopedia of Genes and Genomes (KEGG) enrichment analysis of differentially expressed genes (DEGs) in apple in the comparisons of 35 °C vs. 25 °C at T1, T2, T3, and T4. T1, T2, T3, and T4 represent apples after 3, 6, 9, and 12 days of 25 °C and 35 °C treatment, respectively.

**Figure 5 ijms-23-03766-f005:**
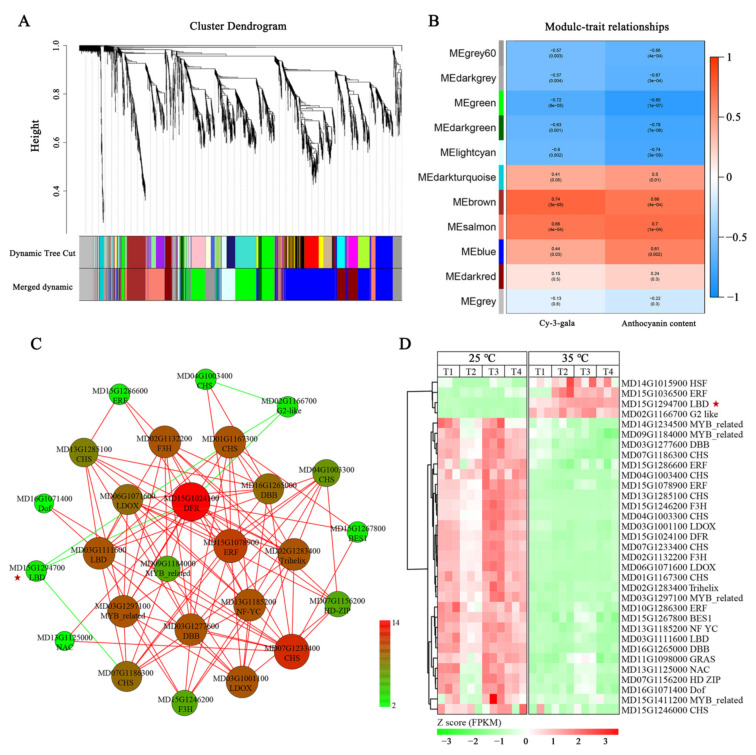
Weighted gene co-expression network analysis (WGCNA) of DEGs in apple at 25 °C and 35 °C treatment. (**A**) Hierarchical cluster tree presenting modules of co-expressed genes with color annotation. (**B**) Module–anthocyanin correlations and corresponding *p*-values. The left panel shows eleven modules. The color scale on the right displays module-trait correlation from −1 (blue) to 1 (red). (**C**) Connection network between 12 anthocyanin structural genes and 21 TFs (transcript factors) in the ‘green’ module. The coefficients were calculated from log2 (fold change) of each transcript using the EXCEL program. This study selected correlations with a coefficient of R2 > 0.9. Positive correlations are presented with red lines, while negative correlations are presented with green lines. The color scale on the right shows anthocyanin structural gene-TF correlation numbers from 2 (green) to 14 (red). The correlation relationships were visualized using Cytoscape (version 2.8.2). (**D**) The heat map for the expression of 12 anthocyanin structural genes and 21 TFs (transcript factors) in the ‘green’ module. The numerical values for the green-to-red gradient bar indicate the log2-fold change relative to the control sample. The red pentagram indicates *MdLBD37*.

**Figure 6 ijms-23-03766-f006:**
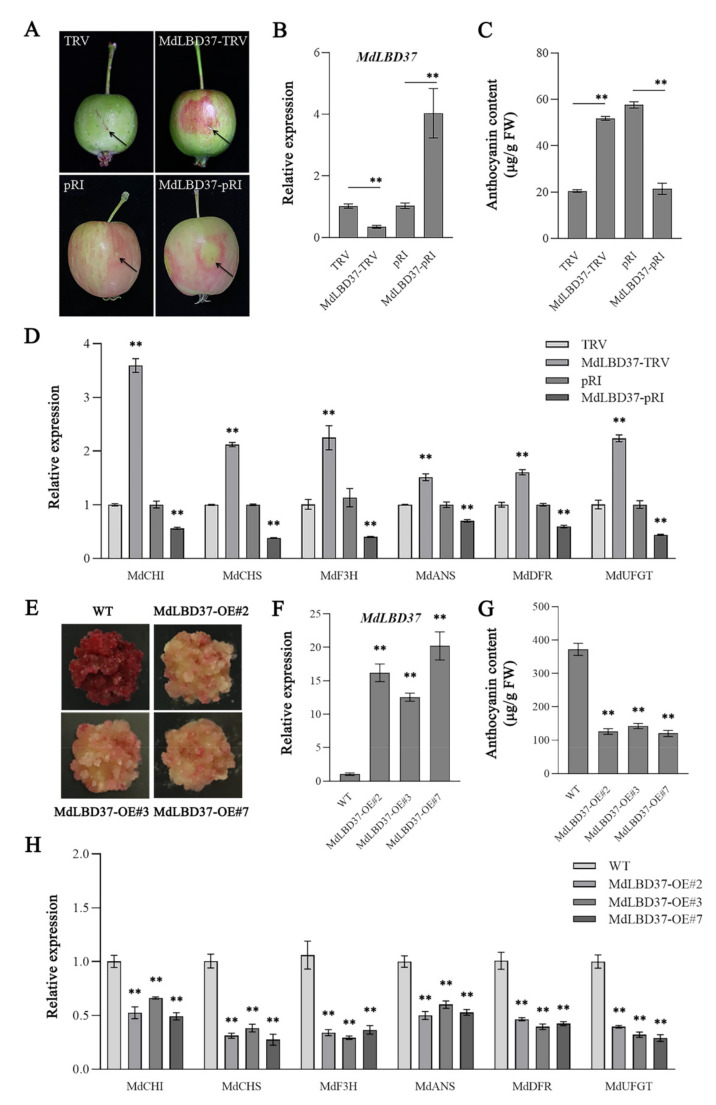
*MdLBD37*-mediated inhibition of anthocyanin synthesis in apple. (**A**) The phenotypes of WT, *MdLBD37*-TRV, and *MdLBD37*-pRI apple fruit. (**B**) The expression levels of *MdLBD37* in WT, *MdLBD37*-TRV, and *MdLBD37*-pRI apple fruit. (**C**) Anthocyanin contents in WT, *MdLBD37*-TRV, and *MdLBD37*-pRI apple fruit. FW, fresh weight. (**D**) The expression levels of anthocyanin-related genes in WT, *MdLBD37*-TRV, and *MdLBD37*-pRI apple fruit. (**E**) Phenotypes of *MdLBD37*-overexpressing (*MdLBD37*-OE) apple calli. (**F**) The expression levels of *MdLBD37* in WT and *MdLBD37*-OE apple calli. (**G**) Anthocyanin contents in WT and *MdLBD37*-OE apple calli. (**H**) The expression levels of anthocyanin-related genes in WT and *MdLBD37*-OE apple calli. FW, fresh weight. Values suggest means ± SD of three independent biological replicates. Statistical significance was computed with the use of Student’s *t*-test (** *p* < 0.01).

**Figure 7 ijms-23-03766-f007:**
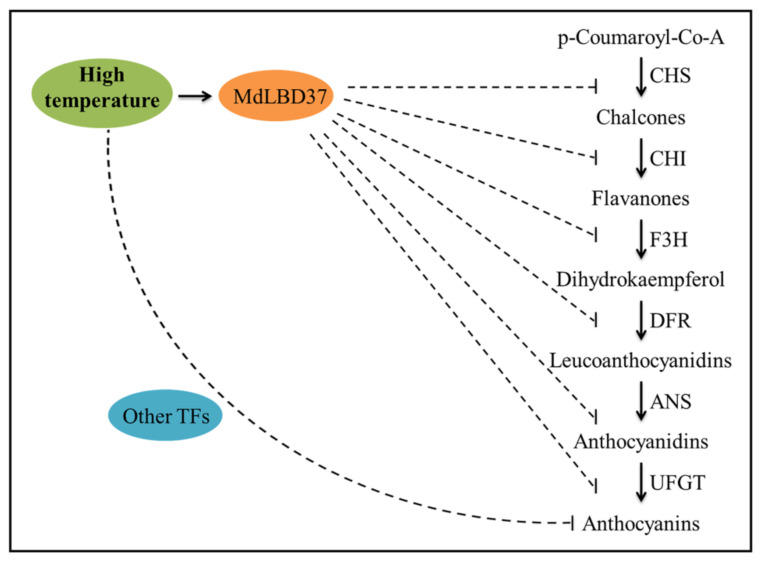
A hypothetical model depicting MdLBD37 and other TFs from the WGCNA module ‘green’ might be involved into the inhibition of high temperature on anthocyanin synthesis. The dotted line indicated possible links.

## Data Availability

The data presented in this study are available on request from the corresponding author.

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
