# Peer review of "Transcriptome Analysis of Apples in High-Temperature Treatments Reveals a Role of MdLBD37 in the Inhibition of Anthocyanin Accumulation"

_ijms, 2022, doi:10.3390/ijms23073766_

Round 1
Reviewer 1 Report
In this manuscript, the author did transcriptome analysis of apple in high-temperature treatments reveals a role of MdLBD37 in the inhibition of anthocyanin accumulation. In this study, the transcriptomes of 25 °C-and 35 °C-treated apples were analyzed by high-throughput RNA sequencing. A total of 8354 differentially expressed genes (DEGs) were detected at four-time points corresponding to the two temperature treatments. The up-regulated DEGs were annotated using GO as well as KEGG databases. A network module of 528 genes (including 21 transcription factors) most associated with the total anthocyanin and cyanidin 3-galactoside contents was constructed by weighted correlation network analysis (WGCNA). In the WGCNA module, we unearthed a LOB domain-containing protein designated as MdLBD37. The expression of MdLBD37 was sharply up-regulated by high temperature and negatively correlated with the total anthocyanin and cyanidin 3-galactoside contents. Authors found that MdLBD37 negatively regulated the expression of anthocyanin biosynthetic genes MdCHI, MdCHS, MdF3H, MdANS, MdDFR, and MdUFGT, as well as anthocyanin accumulation. These results suggested that MdLBD37 significantly influenced the high-temperature inhibition of anthocyanin accumulation in apples. The findings shed more light on the mechanism of anthocyanin inhibition during high-temperature stress in apples.
The manuscript is written well. However, for the betterment of this manuscript, I have a few suggestions and queries for the authors.
- How did the author decide that MdLBD37 negatively regulated the expression of anthocyanin biosynthetic genes MdCHI, MdCHS, MdF3H, MdANS, MdDFR, and MdUFGT, as well as anthocyanin accumulation? Did the author make Ox lines? If yes, please explain the vector as a map with promoter and terminator and the enzyme sites. This experiment is not conclusive author may need to check if MdLBD37 really interacts with anthocyanin biosynthetic genes MdCHI, MdCHS, MdF3H, MdANS, MdDFR, and MdUFGT.
- Make one hypothetical figure depicting the finding of this study.
- To conclude this finding, the authors need to functionally characterize atleat one gene found in this study as OX or KO.
Author Response
Dear Editor,
Thank you for your response and the reviewers’ comments regarding our manuscript, ‘Transcriptome Analysis of Apple in High-Temperature Treatments Reveals a Role of MdLBD37 in the Inhibition of Anthocyanin Accumulation’ (ijms-1637833). We carefully reviewed these valuable comments and revised the manuscript accordingly. We hope you and the reviewers will be satisfied with our changes. Our responses to the comments are provided below.
Respond to Reviewer #1’ comments
- How did the author decide that MdLBD37 negatively regulated the expression of anthocyanin biosynthetic genes MdCHI, MdCHS, MdF3H, MdANS, MdDFR, and MdUFGT, as well as anthocyanin accumulation? Did the author make Ox lines? If yes, please explain the vector as a map with promoter and terminator and the enzyme sites. This experiment is not conclusive author may need to check if MdLBD37 really interacts with anthocyanin biosynthetic genes MdCHI, MdCHS, MdF3H, MdANS, MdDFR, and MdUFGT.
Response: Thank you for your constructive comments. We have analyzed the expression of anthocyanin biosynthetic genes MdCHI, MdCHS, MdF3H, MdANS, MdDFR, and MdUFGT, and anthocyanin content in MdLBD37-overexpressing (MdLBD37-OE) apple calli (Fig 6E-H). We found that overexpression of MdLBD37 in apple calli lowered the expression of MdCHI, MdCHS, MdF3H, MdANS, MdDFR, and MdUFGT, as well as anthocyanin accumulation. Thus, we functionally characterized MdLBD37 as a negative anthocyanin regulator. The interaction between MdLBD37 and anthocyanin biosynthetic genes MdCHI, MdCHS, MdF3H, MdANS, MdDFR, and MdUFGT cannot be concluded based on our present study. However, we believe that these results are sufficient for the following conclusion: 'MdLBD37 represses anthocyanin synthesis possibly by inhibiting anthocyanin biosynthetic genes.' This has been toned down from our original conclusion that 'MdLBD37 negatively regulated the expression of anthocyanin biosynthetic genes MdCHI, MdCHS, MdF3H, MdANS, MdDFR, and MdUFGT, as well as anthocyanin accumulation.'
- Make one hypothetical figure depicting the finding of this study.
Response: Done (Fig 7).
- To conclude this finding, the authors need to functionally characterize atleat one gene found in this study as OX or KO.
Response: Thank you for your constructive comments. We have functionally characterized MdLBD37 in MdLBD37-overexpressing (MdLBD37-OE) apple calli (Fig 6E-H).
In addition, the manuscript has been revised and re-polished by native English speakers.
We have improved the manuscript by addressing the reviewer comments and suggestions. These changes have not modified the content and framework of the manuscript. All changes are marked in yellow in the revised manuscript.
Once again, thank you very much for your comments and suggestions.
Sincerely,
Shouqian Feng, Xuesen Chen
College of Horticulture Science and Engineering
Shandong Agricultural University

Reviewer 2 Report
The manuscript by Yu-Feng et al. reports very interesting data. The anthocyanin content of a variety is mainly determined by genetic factors, but environmental factors, are other important regulators. The unraveling foundation of high-temperature inhibition of apple coloring is essential.
However, some issues need to be clarified or supplemented. The comments are included below.
Picture 1.
- Why is the Cy-3-galactoside content expressed in μg / ml and not in μg / g?
- The symbols in the figures A, B, C do not agree with the description under the figure.
Picture nr 6.
- The symbols in the figures A, B, C, D do not agree with the description under the figure.
- Did a single extraction allow complete extraction of the anthocyanins? Multiple extractions are required to isolate all anthocyanins. How do the authors justify the adoption of such a research method?
-Why was the total anthocyanin content determined by spectrophotometry and not by HPLC. The HPLC method is more accurate.
- Was 1% (v / v) HCl-methanol sufficient for complete anthocyanin extraction? Usually, a higher concentration of methanol is used. How do the authors justify the adoption of such a research method?
4.2. Anthocyanin assessment
4.3. RNA-seq and data analysis
- Total RNA was extracted using TRIzol reagent (Invitrogen, USA) as per specific protocols. This was followed by cDNA preparation based on high-quality RNA. - What protocols? Please specify more precisely.
Author Response
Dear Editor,
Thank you for your response and the reviewers’ comments regarding our manuscript, ‘Transcriptome Analysis of Apple in High-Temperature Treatments Reveals a Role of MdLBD37 in the Inhibition of Anthocyanin Accumulation’ (ijms-1637833). We carefully reviewed these valuable comments and revised the manuscript accordingly. We hope you and the reviewers will be satisfied with our changes. Our responses to the comments are provided below.
Respond to Reviewer #2’ comments
- Picture 1.
- Why is the Cy-3-galactoside content expressed in μg / ml and not in μg / g?
Response: Thank you for your constructive comments. The Cy-3-galactoside content has been expressed in μg / g.
- The symbols in the figures A, B, C do not agree with the description under the figure.
Response: Corrected (Fig 1A-C).
- Picture 6.
- The symbols in the figures A, B, C, D do not agree with the description under the figure.
Response: Corrected (Fig 6A-D).
- Anthocyanin assessment
- Did a single extraction allow complete extraction of the anthocyanins? Multiple extractions are required to isolate all anthocyanins. How do the authors justify the adoption of such a research method?
- Was 1% (v / v) HCl-methanol sufficient for complete anthocyanin extraction? Usually, a higher concentration of methanol is used. How do the authors justify the adoption of such a research method?
Response: Thank you for your constructive comments. In this study, we extracted total anthocyanin from apple samples using 1% (v / v) HCl-methanol extraction, which has been widely used for extracting total anthocyanin in apple, radish and raspberry (An et al., Plant and Cell Physiology, 2015, 56: 650-662; Wang et al., Journal of Experimental Botany, 2021, 72: 6382-6399; Wu et al., Scientific Reports, 2016, 6: 29164; Li et al., Royal Society of Chemistry, 2021, 11: 10804-10813).
-Why was the total anthocyanin content determined by spectrophotometry and not by HPLC. The HPLC method is more accurate.
Response: Thank you for your constructive comments. In this study, total anthocyanin content was determined by spectrophotometry and calculated using pH differential method, which has been widely used for anthocyanin content assessment (Zhang et al., Plant Molecular Biology, 2018, 98: 205-218; Wang et al., The Plant Journal, 2018, 96:39-55; Wang et al., Journal of Experimental Botany, 2021, 72: 6382-6399). In this study, cyanidin 3-galactoside content was identified by HPLC. Related statements have been added to '4.2. Anthocyanin assessment' in 'Materials and Methods' section.
- RNA-seq and data analysis
- Total RNA was extracted using TRIzol reagent (Invitrogen, USA) as per specific protocols. This was followed by cDNA preparation based on high-quality RNA. - What protocols? Please specify more precisely.
Response: We have rewritten this section.
In addition, the manuscript has been revised and re-polished by native English speakers.
We have improved the manuscript by addressing the reviewer comments and suggestions. These changes have not modified the content and framework of the manuscript. All changes are marked in yellow in the revised manuscript.
Once again, thank you very much for your comments and suggestions.
Sincerely,
Shouqian Feng, Xuesen Chen
College of Horticulture Science and Engineering
Shandong Agricultural University

Round 2
Reviewer 1 Report
I am more than happy with the author's response and think that manuscript can be accepted after a small correction.
In figure 2C in expression analysis, the gene name or id should be italic.
Author Response
Respond to Reviewer #1’ comments
1. In figure 2C in expression analysis, the gene name or id should be italic.
Response: Corrected (Fig 2C).
Thank you very much for your comments and suggestions.
Sincerely,
Shouqian Feng, Xuesen Chen
College of Horticulture Science and Engineering
Shandong Agricultural University
Reviewer 2 Report
The article has been sufficiently improved.
Author Response
Thank you very much for your comments and suggestions.
Sincerely,
Shouqian Feng, Xuesen Chen
College of Horticulture Science and Engineering
Shandong Agricultural University